# Multi-Granularity Information Interaction Framework for Incomplete Utterance Rewriting

**Haowei Du [1,2,3,*], Dingyu Zhang[3,*], Chen Li [3], Yang Li[3]**
**Dongyan Zhao[1,2,†]**

[1]Wangxuan Institute of Computer Technology, Peking University
[2] Institute for Artificial Intelligence, Peking University [3] Ant Group
duhaowei@stu.pku.edu.cn, zhangdinghao.zdh@antgroup.com
zhaodongyan@pku.edu.cn, wenyou.lc@antgroup.com, ly200170@alibaba-inc.com

## Abstract

Recent approaches in Incomplete Utterance Rewriting (IUR) fail to capture the source of important words, which is crucial to edit the incomplete utterance, and introduce words from irrelevant utterances. We propose a novel and effective multi-task information interaction framework including context selection, edit matrix construction, and relevance merging to capture the multi-granularity of semantic information. Benefiting from fetching the relevant utterance and figuring out the important words, our approach outperforms existing state-of-the-art models on two benchmark datasets Restoration-200K and CANAND in this field.

## 1 Introduction

Recently increasing attention has been paid to multi-turn dialogue modeling (Choi et al., 2018; Reddy et al., 2019; Sun et al., 2019) and the major challenge in this field is that speakers tend to use incomplete utterances for brevity, such as referring back to (i.e., co-reference) or omitting (i.e., ellipsis) entities or concepts that appear in dialogue history. Su et al. (2019) shows that ellipsis and co-reference can exist in more than 70% of dialogue utterances. To handle this, the task of Incomplete Utterance Rewriting (IUR) (Pan et al., 2019; Elgohary et al., 2019) is proposed to rewrite an incomplete utterance into an utterance which is semantically equivalent but self-contained to be understood without context.

To maintain the similarity of semantic structure between the incomplete utterance and rewritten utterance, recent approaches formulate it as a word edit task (Liu et al., 2020; Zhang et al., 2022) and predict the edit types by capturing the semantic relations between words. However, the sentence-level semantic relations between contextual utterances and incomplete utterance are ne-

| Turns | Utterance |
|---|---|
| $u_1$ | Parsons studied biology at Amherst college. |
| $u_2$ | Who is one of his professors at Amherst? |
| $u_3$ | Parsons ' biology professors at Amherst were Glaser and Henry. |
| $u_4$ | What are his interest? |
| $u_5$ | Parsons showed from early on, a great interest in the topic of philosophy, |
| $u_6$ | Anything else he was interested to? |
| $u_6^*$ | Other than philosophy, is there anything else parsons was interested in? |
| RUN | Besides biology Glaser and Henry, anything else was he interested to? |

Table 1: One example from CANARD dataset. $u_1$-$u_5$ are 5 turns of contextual utterances, $u_6$ denotes the incomplete utterance, $u_6^*$ denote the golden rewriten utterance, and "RUN" denotes the incorrect result of baseline (Liu et al., 2020).

glected. Unaware of which sentences contain the words needed to rewrite the incomplete utterance (important words)(Inoue et al., 2022), these models introduce incorrect words from irrelevant sentences into the rewritten utterance.

We take an example in table 1. The incomplete utterance has the phenomenon of co-reference ("he") and ellipsis ("anything else"). Because the baseline model RUN (Liu et al., 2020) does not fetch the correct source sentence ($u_5$) and the important words ("philosophy"), it introduces irrelevant words ("Glaser and Henry") into rewriting the utterance.

To identify the source sentences of important words and incorporate the sentence-level relations among contextual utterances and incomplete utterance, we propose our multi-granularity information capturing framework for IUR. Firstly, we classify the sentences in contexts into relevant or irrelevant

---

[*]These authors contributed equally to this work.
[†]Corresponding Author.

utterances and match the rewritten utterance with the relevant contexts. Then we capture the token-level semantic relations to construct the token edit matrix. The predicted relevances among sentences in contexts and incomplete utterance, which encodes the sentence-level relations, are utilized to mask the edit matrix. The rewritten utterance is derived by referring to the token matrix. We conduct experiments on two benchmark datasets in IUR and outperform the prior state-of-the-art by about 1.0 score in Restoration-200K dataset and derive competitive performance in CANARD dataset across different metrics including BLEU, ROUGE and F-score.

Our contributions can be summarized as:

**1.** We are the first to incorporate sentence-level semantic relations between the utterances in contexts and the incomplete utterance to enhance the ability to seize the source sentence and figure out the important words.

**2.** We propose the multi-task information interaction framework to capture the multi-granularity of semantic information. Our approach outperforms existing methods on the benchmark dataset of this field, becoming the new state-of-the-art.

## 2 Related Work

There are two main streams of approaches to tackle the task of IUR: generation-based (Huang et al., 2021; Inoue et al., 2022) and edit-based (Liu et al., 2020; Si et al., 2022). Generation-based models solve this task as a seq2seq problem. Su et al. (2019) utilize pointer network to respectively predict the prob of tokens in rewritten utterance from contexts and incomplete utterance. Hao et al. (2021) formulate the task as sequence tagging to reduce the search space. Huang et al. (2021) combine a source sequence tagger with an LSTM-based decoder to maintain the grammatical correctness. Generation models lack to capture the trait of IUR, where the main semantic structure of a rewritten utterance is usually similar to the original incomplete utterance.

Edit-based models focus on predicting word-level or span-level edit type between contextual utterances and the incomplete utterance. Liu et al. (2020) formulate this task as semantic segmentation and propose U-Net (Ronneberger et al., 2015; Oktay et al., 2018) to encode the word-level semantic relations. To incorporate the rich information in self-attention weights of pretrained language model

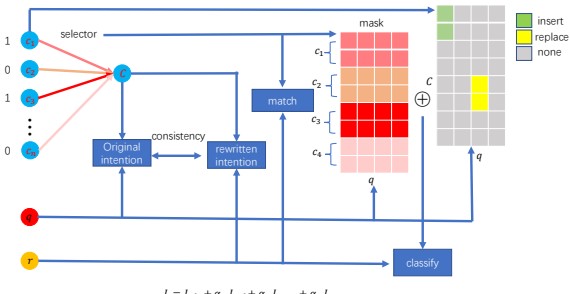

Figure 1: Model pipeline. Our model contains 4 parts: context selection, edit matrix construction, relevance merging and intention check.

(PTM) (Devlin et al., 2018), Zhang et al. (2022) directly take the self-attention weight matrix as the word2word edit matrix. Though these models produce competitive performance, the sentence-level semantic relations are neglected and the models tend to introduce incorrect words from irrelevant contextual utterances.

## 3 Methodology

### 3.1 Overview

By figure 1, our approach contains four components: context selection, edit matrix construction, relevance merging and intention check.

### 3.2 Context Selection

In this part, we capture the semantic relations between contextual utterances and incomplete utterance. Following RUN, We utilize BERT to encode the contextual representation of contexts and incomplete utterance. The input to PTM is the sequence of contexts concatenated by incomplete utterance, and the "[SEP]" token is applied to separate different sentences. The utterance representation is derived by pooling the hidden states of words it contains: $[\mathbf{c_1}, \mathbf{c_2}, \cdots, \mathbf{c_n}, \mathbf{u}] = \mathbf{Pooling}(\mathbf{BERT}[\mathbf{w_1}, \mathbf{w_1}, \cdots, \mathbf{w_N}])$, where $\mathbf{c_i}$ denotes the representation of i-th utterance in contexts, $u$ denotes the representation of incomplete utterance and $w_i$ denotes the i-th word token in the input sequence. We apply a MLP classifier to predict the relevance between each utterance in contexts and incomplete utterance:

$$L_{sel} = \mathbf{CrossEntropy}(\{r_i\}, \{R_i\}) \quad (1)$$
$$r_i = \mathbf{MLP}([\mathbf{c_i}; \mathbf{u}]) \quad i = 1, 2, \cdots, n \quad (2)$$

where $r_i$ and $R_i$ denote the predicted relevance and label of i-th utterance in contexts respectively.

| Model | B1 | B2 | R1 | R2 | F1 | F2 | F3 |
|-------|-----|-----|-----|-----|------|------|------|
| SARG | 92.2 | 89.6 | 92.1 | 86.0 | 62.4 | 52.5 | 46.3 |
| RAST | 90.4 | 89.6 | 91.2 | 84.3 | - | - | - |
| RUN | 92.3 | 89.6 | 92.4 | 85.1 | 68.6 | 56.0 | 47.7 |
| RAU | 92.4 | 89.6 | 92.8 | 86.0 | 69.9 | 57.5 | 49.6 |
| QUEEN | 92.4 | 89.8 | 92.5 | 86.3 | - | - | - |
| Ours | **93.1** | **90.4** | **93.2** | **86.6** | **70.8** | **58.5** | **50.5** |

Table 2: Evaluation results on Restoration-200K dataset. "B1", "B2", "R1", "R2" denote BLEU1, BLEU2, ROUGE1, ROUGE2 respectively. The p-values of ROUGE, BLEU and F-score are smaller than 0.001.

| Model | B1 | B2 | R1 | RL |
|-------|-----|-----|-----|-----|
| RAST | 53.5 | 47.6 | 62.7 | 61.9 |
| RUN | 70.5 | 61.2 | 79.1 | 74.7 |
| HCT | 68.7 | 62.3 | 80.0 | 79.4 |
| QUEEN | 72.4 | **65.2** | **82.5** | **81.8** |
| ChatGPT | 61.5 | 52.5 | 46.1 | 40.1 |
| Ours | **72.9** | 63.2 | 79.2 | 77.0 |

Table 3: Evaluation results on CANARD dataset. "B1", "B2", "R1", "RL" denote BLEU1, BLEU2, ROUGE1, ROUGEL respectively. The baseline results are from original papers.

Considering the golden label is not provided, we retrieve the utterance that contains the words used to rewrite the utterance as the label of relevant utterance. To enhance the compatibility of selected contextual utterance with the rewritten utterance, we sample negative utterances from contexts of other cases in the batch and predict the match score:

$$L_{mat} = \mathbf{CrossEntropy}(\{m_i\}, \{M_i\}) \quad (3)$$

$$m_i = \mathbf{MLP}([\mathbf{c_i} + \mathbf{u}; \mathbf{r}]) \quad i \in C_P \cup C_N \quad (4)$$

$$M_i = \begin{cases} 1 & i \in C_P \\ 0 & i \in C_N \end{cases} \quad (5)$$

where $m_i$ and $M_i$ denote the predicted and labeled matching score of i-th contextual utterance, $C_P$ and $C_N$ denote the golden contextual utterance that includes important words and the sampled negative utterance.

### 3.3 Edit Matrix Construction

Following (Liu et al., 2020), we predict to-ken2token edit type (**Insert**, **Replace**, **None**) based on word representations from PTM with U-Net and build the word edit matrix. The entry at row i and column j in the matrix denotes the edit type between i-th token in contexts and j-th token in incomplete utterance. It can be formulated as:

$$\mathbf{e_{ij}} = \mathbf{MLP}(U(\mathbf{w_i}, \mathbf{w_j})) \quad (6)$$

where $e_{ij} \in \mathrm{R}^3$ denotes the predicted probability of 3 edit types between i-th token in contexts and j-th token in incomplete utterance, $1 \leq i \leq N_C, 1 \leq j \leq N_U$, $U$ denotes the U-Net architecture, $N_C$ and $N_U$ denotes the context length and incomplete utterance length.

### 3.4 Relevance Merging

If some utterance in contexts is classified as relevant by our Context Selection module, it is more possible for the words in this utterance to be adopted into editing the incomplete utterance. In this part, the relevant confidence of the utterance is equally added to the predicted prob of edit type **Insert** and **Replace** for all its constitutive words with the words in incomplete utterance:

$$L_{edit} = \mathbf{CrossEntropy}(\{\hat{\mathbf{e}}_{\mathbf{ij}}\}, \{\mathbf{E_{ij}}\}) \quad (7)$$

$$\hat{e}_{ij}^{Insert} = e_{ij}^{Insert} + \alpha * r_i \quad (8)$$

$$\hat{e}_{ij}^{Replace} = e_{ij}^{Replace} + (1 - \alpha) * r_i \quad (9)$$

where $\mathbf{E_{ij}}$ denotes the golden edit type between i-th token in contexts and j-th token in incomplete utterance and $\alpha$ denotes parameters to tune. The relevance merging process can be seen as utilizing the relevance predicted to "softly mask" the edit matrix. Even if one sentence is not selected as relevant contexts, the probabilities of edit types **Insert** and **Replace** are not strictly set to zero.

### 3.5 Intention Check

To maintain the intention consistency between the incomplete utterance and the rewritten utterance, we project the representation of incomplete utterance and rewritten utterance into intention space

| Model | B1 | B2 | R1 | R2 | F1 | F2 | F3 |
|---|---|---|---|---|---|---|---|
| RUN | 92.3 | 89.6 | 92.4 | 85.1 | 68.6 | 56.0 | 47.7 |
| +cs | 92.4 | 89.5 | 93.2 | 86.3 | 70.2 | 57.1 | 48.8 |
| +cs/hm | 92.8 | 90.2 | 93.3 | 86.7 | 68.6 | 56.4 | 48.6 |
| +cs/sm | 93.0 | 90.3 | 93.4 | **86.8** | 69.1 | 56.7 | 48.7 |
| +cs/sm/ic | **93.2** | **90.5** | **93.4** | 86.7 | 70.0 | 57.6 | 49.7 |
| Ours (+cs/sm/ic/cm) | 93.1 | 90.4 | 93.2 | 86.6 | **70.8** | **58.5** | **50.5** |

Table 4: Ablation results on Restoration-200K dataset. "B1", "B2", "R1", "R2" denote BLEU1, BLEU2, ROUGE1, ROUGE2. "cs", "cm", "sm", "hm", "ic" denote context selection, context marching, relevance soft mask, relevance hard mask, intention checking modules respectively.

and close the distance.

$$L_{int} = \mathbf{KL}(\mathbf{MLP}([\mathbf{C}; \mathbf{u}]), \mathbf{MLP}(\mathbf{r})) \quad (10)$$

$$C = \mathbf{Pooling}(\{\mathbf{c_i}, i \in C_P\}) \quad (11)$$

where $C$ denotes the pooling representations of utterances in contexts and $KL$ denotes the KL-divergence function.

### 3.6 Training and Rewriting

The final training loss function is computed by the weighted sum of edit loss, selection loss, matching loss and intention loss:

$$L_{final} = L_{edit} + \alpha_1 L_{sel} + \alpha_2 L_{mat} + \alpha_3 L_{int}$$

where $\alpha_i, i = 1, 2, 3$ denote parameters to tune. The rewritten utterance is manipulated based on the predicted edit matrix $[\hat{e}_{ij}]_{1 \le i \le N_C}^{1 \le j \le N_U}$. That is, if $\hat{e}_{ij} = \mathbf{Insert}$, we insert the i-th token in contexts before the j-th token in incomplete utterance, which is similar for **Replace** type.

## 4 Experiments

### 4.1 Experimental Setup

**Datasets**: We do experiments on two benchmark IUR datasets from different languages: Restoration-200K (Chinese, (Pan et al., 2019)) and CANARD (English,(Elgohary et al., 2019)). **Metrics**: Following Liu et al. (2020), we utilize BLEU (Papineni et al., 2002), ROUGE (Lin, 2004) and F-score (Pan et al., 2019) as evaluation metrics.
**Baselines**: we compare our approach with recent competitive approaches as follows: RUN (Liu et al., 2020) formulates IUR as a semantic segmentation task and obtains a much faster inference speed than generating from scratch, which can be seen as our backbone; SARG (Huang et al., 2021) proposes a semi auto-regressive generator with the high efficiency and flexibility; RAU (Zhang et al., 2022)

directly extracts the co-reference and omission relationship from the self-attention weight matrix of the transformer; RAST (Hao et al., 2021) proposes a novel sequence-tagging based model to reduce the search space; QUEEN (Si et al., 2022) designs an explicit query template to bring guided semantic structural knowledge; HCT (Jin et al., 2022) constructs a hierarchical context tagger that mitigates the multiple spans issue by predicting slotted rules.

### 4.2 Results

By table 2 and 3, compared with our backbone RUN, our model improves about 1.0 ROUGE and BLEU score, and 2.5 F-score in Restoration-200K dataset, as well as 2.0 ROUGE and BLEU score in CANARD dataset. It demonstrates the efficiency of our multi-task framework to fetch the source of important words and avoid irrelevant words. Specially, we outperform QUEEN by 0.7 BLEU1, 0.6 BLEU2, 0.7 ROUGE1 and 0.3 ROUGE2, and beat RAU by about 1.0 F-score on Restoration-200K dataset, becoming the new state-of-the-art. Moreover, our model shows competitive performance on CANARD dataset, which beats QUEEN by 0.5 BLEU1 score.

### 4.3 Ablation Study

To explore different modules of our approach, we conduct the ablation study. Compared with the "soft mask" of relevance merging in section 3.4, we design an ablation with "hard mask" merging: if a sentence in contexts is classified as irrelevant in section 3.2, the probabilities of **Insert** and **Rewrite** between its words and the words in the incomplete utterance are set to 0. In table 4, context selection, relevance merging and intention checking show progressive improvement across different metrics. Compared with hard mask of relevance merging, the soft mask method is overall better. The sentence classified as irrelevant may still contain im-

portant words, so merging its relevance in a soft way is necessary. Context marching module helps the approach to capture the important words from contexts.

## 5 Conclusion

In this paper, we argue that capturing the source of important words can diminish to introduce irrelevant words for IUR. We propose a novel and effective multi-task framework including context selection, edit matrix construction, and relevance merging to capture the multi-granularity of semantic information and fetch the relevant utterance. we do experiments on two benchmark datasets in IUR and show competitive performance.

## Limitations

We propose a novel and effective multi-task framework to capture the multi-granularity of semantic information and fetch the relevant utterance. With the population of large language model (LLM), the multi-task finetuning framework may bring more computation cost. We will explore the combination of our approach with LLM in the future.

## Acknowledgements

This work was supported by the National Key Research and Development Program of China (No.2021YFC3340304).

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

# A   Example Appendix

This is a section in the appendix.