# OpenReview forum: "Multi-Granularity Information Interaction Framework for Incomplete Utterance Rewriting"
_EMNLP/2023/Conference — EMNLP 2023 Findings_

### Official Review · Reviewer_VzV8 · 2023-08-09

**Soundness:** 3

**Excitement:**

3: Ambivalent: It has merits (e.g., it reports state-of-the-art results, the idea is nice), but there are key weaknesses (e.g., it describes incremental work), and it can significantly benefit from another round of revision. However, I won't object to accepting it if my co-reviewers champion it.

**Paper Topic And Main Contributions:**

The authors introduced an innovative and efficient framework for multi-task information interaction, encompassing context selection, edit matrix construction, and relevance merging to capture the multi-granularity of semantic information. The method surpasses current state-of-the-art models on two established benchmark datasets, namely Restoration-200K and CANAND.

**Questions For The Authors:**

1. What are the model sizes for different components? How efficient is this system in terms of computational cost/latency?

**Reasons To Accept:**

1. The chosen methodology demonstrates a clear and relevant motivation, effectively aligning itself with the objectives of the Incomplete Utterance Rewriting (IUR) task. This indicates a thoughtful understanding of the task's requirements and goals.
2. Its ability to attain SOTA performance proves its efficacy.

**Reasons To Reject:**

1. As mentioned in limitation section, given the advances in generative LLMs, an end-to-end LLM may be preferable, assuming it can yield decent performance. Even though LLM is not a focus of the paper, ideally it should be included as a baseline/benchmark for completeness.

**Reproducibility:**

2: Would be hard pressed to reproduce the results. The contribution depends on data that are simply not available outside the author's institution or consortium; not enough details are provided.

**Reviewer Confidence:**

4: Quite sure. I tried to check the important points carefully. It's unlikely, though conceivable, that I missed something that should affect my ratings.

---

> ### Author Rebuttal · Authors · 2023-08-29
>
> R1: We conduct an experiment using ChatGPT in-context learning ability (ICL). We randomly select 5 examples and append them to the task instruction as the demonstration, followed by rewriting the incomplete utterance. The performance on CANARD is as follows: B1: 61.5, B2: 52.5, R1: 46.1, R2: 40.1. It shows the ICL performance of LLM is significantly lower than IUR methods, and there is still exploration room on fitting the LLM for IUR task. We will add this baseline in revised version.
>
> Q1: There are about 110M parameters in text encoder (BERT), 1M parameters in edit matrix construction (U-net), and 20K parameters in context selection, context matching and intention checking. The latency is about 20ms to produce a single sentence with PyTorch on a single NVIDIA V100. The modules we propose will not significantly improve the model size and latency than the backbone.

---

### Official Review · Reviewer_9o7i · 2023-08-11

**Soundness:** 3

**Excitement:**

3: Ambivalent: It has merits (e.g., it reports state-of-the-art results, the idea is nice), but there are key weaknesses (e.g., it describes incremental work), and it can significantly benefit from another round of revision. However, I won't object to accepting it if my co-reviewers champion it.

**Paper Topic And Main Contributions:**

This paper introduces a multi-task knowledge interaction architecture to  capture multi-granularity of semantic information (sentences and words) to improve utterance infilling (e.g, ellipsis and co-reference) task.

**Questions For The Authors:**

1. Does the proposed method  also work for rewriting  dialog state information in incomplete utterances ( the pair of  slot type and slot value)?

**Reasons To Accept:**

1. This paper raises a important issue for incomplete utterance rewriting of multi-turn dialogues.

2. This paper propose a novel approach to improve utterance infilling task by considering both utterance-level and word-level  semantic relations.

3.  How the method improve the performance on incomplete utterance infilling of multi-turn dialogues keep basically clear.


**Reasons To Reject:**

1. The proposed method in the paper fails to consider how to rewrite incomplete utterances which refers to dialog state ( the pair of  slot type and slot value) infilling problem.
2.  The paper does not carry any experiments on large language models for incomplete utterance infilling performance comparison .

**Reproducibility:**

4: Could mostly reproduce the results, but there may be some variation because of sample variance or minor variations in their interpretation of the protocol or method.

**Reviewer Confidence:**

3: Pretty sure, but there's a chance I missed something. Although I have a good feel for this area in general, I did not carefully check the paper's details, e.g., the math, experimental design, or novelty.

---

> ### Author Rebuttal · Authors · 2023-08-29
>
> R1: The IUR task aims to rewrite an incomplete utterance into an utterance which is semantically
> equivalent but self-contained to be understood without context. The identification of dialogue state is not explicitly needed.
>
> R2: We conduct an experiment using ChatGPT in-context learning ability (ICL). We randomly select 5 examples and append them to the task instruction as the demonstration, followed by rewriting the incomplete utterance. The performance on CANARD is as follows: B1: 61.5, B2: 52.5, R1: 46.1, R2: 40.1. It shows the ICL performance of LLM is significantly lower than existing IUR methods, and there is still exploration room on fitting the LLM for IUR task. We will add this baseline in revised version.
>
> Q1: We introduce our context selection and matching module into dialogue state tracking with T5
> sequential generation and improve the JGA score [1] from 51.2 to 52.3 on MultiWOZ 2.2 dataset. It shows incorporating sentence-level semantic relations between the utterances in contexts also helps to seize the important dialogue information.
>
> [1] Dialogue State Tracking with a Language Model using Schema-Driven Prompting. Lee et al. EMNLP2021.
>
> [2] Multiwoz-a largescale multi-domain wizard-of-oz dataset for task-oriented dialogue modeling. Budzianowski et al. EMNLP2018.

---

### Official Review · Reviewer_DJDn · 2023-08-11

**Soundness:** 3

**Excitement:**

3: Ambivalent: It has merits (e.g., it reports state-of-the-art results, the idea is nice), but there are key weaknesses (e.g., it describes incremental work), and it can significantly benefit from another round of revision. However, I won't object to accepting it if my co-reviewers champion it.

**Paper Topic And Main Contributions:**

This is a paper in the field of incomplete discourse rewriting. The paper proposes a novel multi-task information interaction framework including context selection, edit matrix construction, relevance merging and intention check to capture the multi-granularity of semantic information. The paper also constructs corresponding comparative experiments and ablation experiments to prove the effectiveness of the proposed method.

**Reasons To Accept:**

(1)The paper has some innovations. The paper proposes a framework including context selection, edit matrix construction, relevance merging and intention check for incomplete utterance rewriting. The training loss combines edit loss, selection loss, matching loss and intention loss.
(2)The paper constructs comparative experiments and ablation experiments to prove the effectiveness of the proposed method.


**Reasons To Reject:**

(1)The paper has some innovations but the innovations are not very prominent. The loss function is computed by the weighted sum of edit loss, selection loss, matching loss and intention loss. There are no outstanding innovations in the model structure.
(2)The evaluation results on the CANARD dataset in Table 3 are not satisfactory.
(3)From the results of the ablation experiments in Table4, the addition of the context marching module brings about a decrease in the ROUGE and BLEU indicators.


**Reproducibility:**

5: Could easily reproduce the results.

**Reviewer Confidence:**

5: Positive that my evaluation is correct. I read the paper very carefully and I am very familiar with related work.

---

> ### Author Rebuttal · Authors · 2023-08-29
>
> R1: We propose a simple and effective model structure to incorporate sentence-level semantic relations between the utterances in contexts and the incomplete utterance to enhance the ability to seize the source sentence and figure out the important words. This structure does not notably enlarge the model size or latency, and significantly improve the IUR performance compared with the backbone.
>
> R2: We outperform all the baselines by different metrics across BLEU, ROUGE, F-score on Restoration-200K, and baselines other than QUEEN on CANARD. QUEEN constructs the query template by the linguistic feature of the specific dataset and it constrains the effectiveness on general datasets or practical data. And our method can be easily generalized to other datasets without explicit template.
>
> R3: Though bringing a slight reduction (0.1%) on BLEU and ROUGE, context matching improves the F-score by about 1.0%. F-score concentrates more on words from the context (important words) which are argued to be harder to copy [1]. So we keep the context matching module in our method.
>
> [1] Improving open-domain dialogue systems via multi-turn incomplete utterance restoration. Pan et al. EMNLP2019.

---

### Meta-Review · Area_Chair_68SZ · 2023-09-19

**Recommendation:** 3

**Metareview:**

The paper introduces a multi-task knowledge interaction architecture to improve performance on the utterance infilling task.

The paper discusses some interesting points regarding incomplete utterance rewriting of multi-turn dialogues
The paper proposes an effective approach and is able to achieve state-of-the-art results.
The included experiments and ablation tests are informative.

The method has limited novelty according to some reviewers.
Some of the results are mixed, while the ablation tests show a decrease when certain components are added.
The paper lacks comparsion to large language models, which may outperform this system, thereby decreasing its relevance to the current research directions.

---

### Decision · Program_Chairs · 2023-10-07

**Decision:**

Accept-Findings

**Comment:**

The paper introduces a multi-task knowledge interaction architecture to improve performance on the utterance infilling task.

The paper discusses some interesting points regarding incomplete utterance rewriting of multi-turn dialogues
The paper proposes an effective approach and is able to achieve state-of-the-art results.
The included experiments and ablation tests are informative.

The method has limited novelty according to some reviewers.
Some of the results are mixed, while the ablation tests show a decrease when certain components are added.
The paper lacks comparsion to large language models, which may outperform this system, thereby decreasing its relevance to the current research directions.